# Synthesis of 2-Substituted Benzo[b]furans/furo-Pyridines Catalyzed by NiCl₂

**Rong Zhou \*, Yanli Ding and Mili Yang**

College of Chemical Engineering, Xinjiang Agricultural University, Urumqi 830052, China;
dingyanli628@163.com (Y.D.); yml18299157312@163.com (M.Y.)
* Correspondence: zrhg@xjau.edu.cn

**Abstract:** The first Ni-catalyzed tandem synthesis of 2-substituted benzo[b]furans/furo-pyridines from 2-halophenols and 1-alkynes was explored under Cu-free and phosphine-free conditions. The protocol was carried out with NiCl₂/5-nitro-1, 10-phenanthroline in DMA (*N*,*N*-dimethylacetamide) at 120 °C. It was found to be simple, cost effective, and have a wide substrate scope. Additionally, the method is compatible with heteroaryl substrates, resulting in the formation of 2-substituted benzo[b]furans/furo-pyridines in reasonable to good yields.

**Keywords:** Ni; benzo[b]furan; coupling-cyclization reaction

---

## 1. Introduction

2-substituted benzo[b]furans/furo-pyridines are important building blocks in biologically active compounds such as anti-inflammation agents and anti-fungal activities (Scheme 1) [1–6]. Due to these excellent properties, the synthesis of these skeletons has become a hot spot in recent years [7,8]. Recently, York, Panli, and co-workers showed the coupling of unsaturated hydrocarbons with heteroaryl and aryl compounds using Pd catalysis [8,9]. Following their work, several methods were developed to synthesize 2-substituted benzo[b]furan/furo-pyridines [10–13]. However, to obtain these skeletons, noble metals (such as Pd and Rh), air sensitive phosphine ligands, or rigorous conditions usually seem to be necessary [14–17].

**Scheme 1.** Some 2-substituted benzo[b]furans/furo-pyridines with biologically active.

Ni is a superior substitution in catalysis reactions because it is less expensive, air stable, and less toxic [18–20]. However, it has not been used in the construction of 2-substituted benzo[b]furans/furo-pyridines except in one heterogeneous example. Wang Lei et al. developed a catalytic system of ultrafine nickel(0) powder (100 nm)/CuI/PPh₃ to synthesize 2-phenylbenzo[b]furan with a 75% yield in 2004 [21]. Herein, we report on an inexpensive, Cu-free, and phosphine-free method for the construction of 2-substituted benzo[b]furans/furo-pyridines by Ni-catalyzed intermolecular cyclization of 2-iodopehenols or 2-bromophenols and 1-alkynes.

## 2. Results and Discussion

For optimization studies of 2-substituted benzo[b]furan synthesis, 2-iodophenol **1a** and phenylacetylene **2a** were chosen as the model substrates (Scheme 2). At first, a series of nickel salts were screened and $NiCl_2$ was found to be the best, affording the most results and the corresponding product **3a** in an 80% yield (Table 1, entry 1). Various bases were screened to identify the optimized condition where weak and organic bases such as $KHCO_3$ and $Et_3N$ (triethylamine) were found to be unsuitable for this transformation (Table 1, entries 10–11, 16–17). However, a strong base like NaOH could promote this cyclization to give the title product with the yield of 80% (Table 1, entry 1). Solvents have a great influence in this reaction, so some solvents with different polarities were selected for this reaction. The effect of the solvent was also studied and the data showed that DMA provided the best result, while others like DMF (*N,N*-dimethylformamide), toluene, $H_2O$, etc. produced lower yields (Table 1, entries 18–23).

**Scheme 2.** The model reaction.

**Table 1.** Optimization of reaction conditions [1].

| Entry | [Ni] | Base | Solvent | Yield/% |
|---|---|---|---|---|
| 1 | $NiCl_2$ | NaOH | DMA | 80 |
| 2 | $NiCl_2$ | NaOH | DMA | 47 [2] |
| 3 | - | NaOH | DMA | 0.8 |
| 4 | $Ni(dppe)Cl_2$ | NaOH | DMA | 26 |
| 5 | $Ni(dppp)Cl_2$ | NaOH | DMA | 16 |
| 6 | $Ni(PPh_3)_2Cl_2$ | NaOH | DMA | 41 |
| 7 | $Ni(PCy_3)_2Cl_2$ | NaOH | DMA | 13 |
| 8 | $Ni(COD)_2Cl_2$ | NaOH | DMA | 28 |
| 9 | $NiSO_4$ | NaOH | DMA | 75 |
| 10 | $NiCl_2$ | $NaHCO_3$ | DMA | 0.6 |
| 11 | $NiCl_2$ | $KHCO_3$ | DMA | 0.7 |
| 12 | $NiCl_2$ | $Na_2CO_3$ | DMA | 2 |
| 13 | $NiCl_2$ | $Cs_2CO_3$ | DMA | 1 |
| 14 | $NiCl_2$ | $K_3PO_4$ | DMA | 26 |
| 15 | $NiCl_2$ | KOH | DMA | 74 |
| 16 | $NiCl_2$ | $Et_3N$ | DMA | 0.1 |
| 17 | $NiCl_2$ | pyridine | DMA | 0.1 |
| 18 | $NiCl_2$ | NaOH | DMF | 56 |
| 19 | $NiCl_2$ | NaOH | 1,4-Dioxane | 0.2 [3] |
| 20 | $NiCl_2$ | NaOH | Toluene | 0.6 [3] |
| 21 | $NiCl_2$ | NaOH | $H_2O$ | - [3] |
| 22 | $NiCl_2$ | NaOH | n-BuOH | 0.1 [3] |
| 23 | $NiCl_2$ | NaOH | t-BuOH | - [4] |

[1] Reaction conditions: **1a** (1 mmol), **2a** (1.2 mmol), Base (2 mmol), [Ni] (0.1 mmol), 5-nitro-1, 10-phenanthroline (0.1 mmol), DMA (2 mL), 120 °C, 20 h, $N_2$; [2] in air; [3] 100 °C; [4] 80 °C.

Ligands play an important role in the metal catalyzed coupling reaction and so the effect of N-ligands was investigated (Scheme 3). It was shown that the diamine ligand 5-nitro-1,10-phenanthroline **L14** was the most effective ligand. The result revealed that a ligand with a rigid skeleton could make the reaction work more smoothly than that with a flexible one. The ligand with large steric hindrance could hinder the process of the coupling-cyclization reaction (**L2–L4**, **L8–L9**, **L13**). Other bidentate N-ligands with a rigid skeleton such as 4-methyl-1,10-phenanthroline

(**L11**), and 1,7-dichloro-1,10-phenanthroline (**L12**) were found to be less efficient for the conversion, with 72% and 62% yields, respectively.

**Scheme 3.** Screen of ligands. Reaction conditions: **1a** (1 mmol), **2a** (1.2 mmol), NiCl$_2$ (0.1 mmol), **L** (0.1 mmol), NaOH (2 mmol), 120 °C, 20 h, GC yield.

Under the optimized reaction conditions, the functional group tolerance of this reaction was explored (Scheme 4). 2-iodophenols with electron-withdrawing and electron-donating groups could all react with phenylacetylene **2a** and obtain the corresponding products (**3a**–**3g**). EWG (electron-withdrawing groups) with more steric hinderance such as 4-*t*-butyl reduced the reactivity and gave a 37% yield (**3b**). Heteroaryl substrates contain N, which has the potential to combine to nickel, thus poisoning the catalyst. Nevertheless, these heteroaryl materials were all completed smoothly in this system (**3c**–**3d**, **3n**–**3x**). It is noteworthy that 2-iodo-6-methylpyridin-3-ol coupled with 5-bromo-3-iodopyridin-2-ol easily during this system, giving 52% and 35% yields of the products (**3c**, **3d**). The alkynes bearing –OCH$_3$, –CH$_3$, –CH$_2$CH$_2$CH$_3$, –COOCH$_3$, –F, and –Br groups all reacted successfully and produced the desired compounds (**3h**–**3n**) in reasonable to good yields (30–75%). 3-ethynylpyridine was also found to be suitable during this one-pot reaction condition (**3n**–**3s**). When reacted with the large steric hindrance material 4-*t*-butyl-2-iodophenol, it produced 3-(5-(*t*-butyl)benzofuran-2-yl)pyridine **3o** with an 89% yield. When 6-methyl-2-iodo-pyridinol was used to cyclize with various 1-alkynes, such as 4-bromo-phenylacetylene, 4-methyl-phenylacetylene, 4-propyl-phenylacetylene, and the corresponding 2-substituted furo-pyridines can be achieved with medium yields (**3t**–**3w**). Since 2-bromophenols are less reactive and not susceptible to this transformation, the Ni catalyzed system has not been reported in the literature with tandem synthesis. Our results show that the procedure works well in the presence of 2-bromophenol, 2-bromo-6-methylpyridin-3-ol, 2-bromo-4-methylphenol, and even 2-bromopyridin-3-ol as starting materials (**3a**, **3c**, **3x**–**3y**).

**Scheme 4.** The exploration of the universality of substrates. Reaction conditions: **1** (1 mmol), **2** (1.2 mmol), NiCl₂ (0.1 mmol), **L14** (0.1 mmol), NaOH (2 mmol), 120 °C, 40 h, isolated yield.

## 3. Materials and Methods

All chemicals were purchased from commercial companies. All were used as received except for some liquid materials that were sensitive to light and moisture (DMA) being purified prior to use. $^1$H NMR ($^1$H Nuclear Magnetic Resonance) and $^{13}$C NMR ($^{13}$C Nuclear Magnetic Resonance) spectra were measured on a VARIAN 400-MR. Mass spectroscopy data of the products were collected with a MS-EI (Mass spectrometry-Electron ionization) instrument. All products were isolated by chromatography on silica gel (300–400 mesh) using petroleum ether (60–90 °C). Compounds described in the literature were characterized by $^1$H NMR and $^{13}$C NMR spectroscopy and compared to the reported data, detailed information in Supplementary Materials.

NiCl₂ (0.1 mmol), Ligand (0.1 mmol), 2-halophenol (1 mmol), 1-alkynes (1.2 mmol), NaOH (2 mmol), and degassed DMA (2 mL) were added successively into a dried Schlenk tube with a magnetic bar under nitrogen. The reaction was performed at 120 °C. At the end of reaction, the solution was cooled to room temperature and water (3 mL) was added. The mixture solution was extracted with ethyl acetate (3 × 3 mL). The organic layer was dried over MgSO₄, then filtered and purified with silica gel chromatography (petroleum ether) to give a corresponding product.

## 4. Conclusions

In conclusion, a novel, and simple route was developed for the synthesis of 2-substituted benzo[b]furans/furo-pyridines via a tandem Sonogashira coupling-cyclization sequence of 2-iodophenols or 2-bromophenols and 1-alkynes catalyzed by Ni with Cu-free and phosphine-free in a reasonable to good yield.

**Supplementary Materials:** The following are available online at http://www.mdpi.com/2073-4344/9/12/1019/s1, Table S1: Optimization of reaction conditions.

**Author Contributions:** Conceptualization, R.Z.; methodology, R.Z.; investigation M.Y.; resources, R.Z.; data curation, Y.D.; writing—original draft preparation, R.Z.; writing—review and editing, R.Z.; visualization, R.Z.; supervision, R.Z.

**Funding:** This research was funded by the Natural Science Foundation of Xinjiang Province (no. 2016D01B18).

**Conflicts of Interest:** The authors declare no conflict of interest.

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
