# Peer review of "Synthesis of 2-Substituted Benzo[b]furans/furo-Pyridines Catalyzed by NiCl2"

_catalysts, doi:10.3390/catal9121019_

Round 1

Reviewer 1 Report

The manuscript is showing the synthesis of polyaryl compound catalyzed by Nickel.

There are some major re-writing needed for this manuscript to be published. Some of them are as follows:

line 2, "The" can be removed from title. line 8, The current abstract is more like an introduction. It should talk about the current work not background. line 14, maybe better to write "... reasonable to good yield."  line 18, introduction needs a major re-write. as an example, I have re-write some of the phrase as a suggestion. line 18 can be re-write as: "2-substituted benzo[b]furanes/furo-pyridines are important building blocks in biologically active compounds such as anti-inflammation and anti-fungal agents [Ref]." line 19, "Recently, York, Panli, and co-workers showed the coupling of unsaturated hydrocarbons with heteroaryl and aryl compounds using Ni and pd catalysis [Ref]. Some anti-inflammation and anti-fungal compounds as an example can be shown as a scheme in introduction. Line 22 and 23, there are reports of polyaryl synthesis that transition metal catalysts were not used such as "J. Am. Chem. Soc. 2014, 136, 8568", make sure you include those in ref [8-10]. Rephrase line 23-25. Line 26 has to be re-phrased. Line 37 through 40 has to be re-phrased. It would be useful if the mechanism of the reaction will be presented as a scheme. line 47, there is an extra space between number and degreeC (°C).  line 47, degC should be written correctly. Like °C not superscript of letter "O". line 48, please avoid using "We". past participle is preferred.  line 51, same as line 47. line 52 and 53, re-write.  line 57, efficient not efficiency. line 57, there should be a comma after yield. Line 58 and 59, re-write. avoid using  "we". line 61-63, re-write. line 64, which substrate you are talking? please mention it. line 67, "... in reasonable to good yields." line 68, re-write. "And" cannot be used as a first word. line 73, "could be got" - grammatical error, re-write. Line 75, re-write. Line 79, same as line 47, line 82, purchased from where? line 95, please get rid of "cost effective". line 98, "... reasonable to good yield".

Author Response

Dear Reviewer,

Thank you for your comments concerning our manuscript entitled “The Synthesis of 2-Substituted Benzo[b]furans/furo-pyridines Catalyzed by NiCl2”. We have studied their comments carefully and have made correction which we hope meet with their approval. Point by point responses to the comments are listed below.

Reviewer1:

Point 1: line 2, "The" can be removed from title.

Response 1: "The" has been removed from title

Point 2: line 8, The current abstract is more like an introduction. It should talk about the current work not background.

Response 2: The sentence about the background has been deleted in abstract.

Point 3: line 14, maybe better to write "... reasonable to good yield."

Response 3: The sentence has been corrected according to the comment.

Point 4: line 18, introduction needs a major re-write.  as an example, I have re-write some of the phrase as a suggestion. line 18 can be re-write as: "2-substituted benzo[b]furanes/furo-pyridines are important building blocks in biologically active compounds such as anti-inflammation and anti-fungal agents [Ref]. "

Response 4: The sentence has been re-write according to the suggestion.

Point 5: line 19, "Recently, York, Panli, and co-workers showed the coupling of unsaturated hydrocarbons with heteroaryl and aryl compounds using Ni and pd catalysis [Ref].

Response 5: The sentence has been corrected according to the comment.

Point 6: Some anti-inflammation and anti-fungal compounds as an example can be shown as a scheme in introduction.

Response 6: Some anti-inflammation and anti-fungal compounds as an example have been shown as a scheme 1 in introduction.

Point 7: Line 22 and 23, there are reports of polyaryl synthesis that transition metal catalysts were not used such as "J. Am. Chem. Soc. 2014, 136, 8568", make sure you include those in ref [8-10].

Response 7: The reports such as "J. Am. Chem. Soc. 2014, 136, 8568" have been included.

Point 8: Rephrase line 23-25. Line 26 has to be re-phrased. Line 37 through 40 has to be re-phrased.

Response 8: line 23-26 and Line 37 through 40 have been re-phrased.    

Point 9: It would be useful if the mechanism of the reaction will be presented as a scheme.

Response 9: The sentence has been rivised according to the comment.

Point 10: line 47, there is an extra space between number and degreeC (°C).  line 47, degC should be written correctly. Like °C not superscript of letter "O".

Response 10: The incorrect have been changed.

Point 11: line 48, please avoid using "We". past participle is preferred.

Response 11: The sentence has been corrected according to the comment.  

Point 12: line 51, same as line 47.

Response 12: The incorrect have been changed.

Point 13: line 52 and 53, re-write.

Response 13: The sentence has been re-write.

Point 14: line 57, efficient not efficiency. line 57, there should be a comma after yield.

Response 14: The incorrect have been revised.

Point 15: Line 58 and 59, re-write. avoid using  "we".

Response 15: The sentence has been re-write.

Point 16: line 61-63, re-write.

Response 16: The sentence has been re-write.

Point 17: line 64, which substrate you are talking? please mention it.

Response 17: The substrates have been added according to the suggestion.

Point 18:  line 67, "... in reasonable to good yields."

Response 18: The mistake has been corrected.

Point 19: line 68, re-write. "And" cannot be used as a first word. 

Response 19: The sentence has been re-write.

Point 20: line 73, "could be got" - grammatical error, re-write. Line 75, re-write.

Response 20: The sentence has been re-write.

Point 21: Line 79, same as line 47. 

Response 21: The mistake has been corrected.

Point 22: line 82, purchased from where?

Response 22: The missing words have been added.

Point 23: line 95, please get rid of "cost effective".

Response 23: The sentences have been revised according to the comments.

Point 24: line 98, "... reasonable to good yield".

Response 24: The mistake has been corrected

Reviewer 2 Report

The paper by Rong Zhou and co-worker describe the synthesis of 2-substituted benzo[b]furans/furopyridines employing Ni catalysis.

The results are interesting enough to be publised. However, Style & Overall presentation as well as typographical and grammatical mistakes should be strongly improved.

Some specific things among many others:

Line 11, page 1: 2-halphenols to 2-halophenols

With Cu-free and phosphine-free ?? “under Cu-free and phosphine-free conditions”?

Lines 18-19, page 1, Check gramma!

Line 29, page 1: 75% yield in 2004

Scheme between lines 37 and 38; Put on top of the table 1 or list as Scheme 1

Line 49, N-lignads to N-ligands

5-nitro-1,10-……phenantroline is missing

Author Response

Dear Reviewer,

Thank you for your comments concerning our manuscript entitled “The Synthesis of 2-Substituted Benzo[b]furans/furo-pyridines Catalyzed by NiCl2”. We have studied their comments carefully and have made correction which we hope meet with their approval. Point by point responses to the comments are listed below.

Reviewer 2:

Point 1:  Line 11, page 1: 2-halphenols to 2-halophenols. With Cu-free and phosphine-free ?? “under Cu-free and phosphine-free conditions”?

Response 1: The mistakes have been corrected according to the comment.

Point 2: Lines 18-19, page 1, Check gramma!

Response 2: The sentence has been re-write according to the suggestion.

Point 3:  Line 29, page 1: 75% yield in 2004.

Response 3: The sentence has been corrected according to the comment.

Point 4: Scheme between lines 37 and 38; Put on top of the table 1 or list as Scheme 1.

Response 4: Scheme has been put on top of the table 1 according to the suggestion.

Point 5:  Line 49, N-lignads to N-ligands.

Response 5: The mistake has been corrected.

Point 6: 5-nitro-1,10-……phenantroline is missing.

Response 6: The missing word has been added according to the suggestion.

Round 2

Reviewer 1 Report

Thanks for the corrections, the following needs correction as well.

Line 11, "... simple, cost effective, and wide substrate scope. Additionally, the method is compatible with heteroaryl substrates resulting in formation of ...".

Line 12, in a reasonable 

Line 17, please remove "and so on".

Line 20, Instead of "After that" you could use "Following their work" 

Line 21, Systems -----> Methods

Line 22, There should be comma after ligands

Line 23, "...Conditions sees necessary [14-17]."

Line 27, "...air stable, AND less toxic property [18-20].

Line 28, Remove "but" and use "However," instead.

Line 29, "Wang Lei and coworkers developed A catalyTIC system."

Line 31, use "in" instead of "with".

Line 37, "...screened AND NiCl2...."

Line 37, "...found to be the best affording the ..."

Line 37, 2a should be 3a, is that right?

Line 38, "...(table1, entry 1)

Line 38, remove "Then"

Line 40: remove like and use "such as"

Line 42 and 43 is not necessary to be here. you can remove that.

Line 44, The effect of solvent has also been studied and the data showed that DMA provide the best result....

Scheme 2 is not providing any information. Please remove that.

Line 53, has been investigated.

Line 54, remove one and use ligand instead,

Line 65, remove "coupling-cyclize" and use "react"

Line 66, EWG with more steric hinderance such as 4-t-butyl reduces the reactivity and gave 37% yield.

Line 79: "... can be achieved instead of "could be get"

Line 80, transform -----> this transformation 

line 81, use " was not reported in the literature" instead of ".. has no literature .."

Line 81, Remove Pleasurably and use " Our results shows ..."

Line 98, "... a novel AND simple..."

Author Response

Thank you for your comments concerning our manuscript entitled “Synthesis of 2-Substituted Benzo[b]furans/furo-pyridines Catalyzed by NiCl2”. We have studied their comments carefully and have made correction which we hope meet with their approval. Point by point responses to the comments are listed below.

Point 1: Line 11, "... simple, cost effective, and wide substrate scope. Additionally, the method is compatible with heteroaryl substrates resulting in formation of ..."

Response 1: The sentence has been corrected according to the comment.

Point 2: Line 12,  in a reasonable.

Response 2: The sentence has been corrected according to the comment.

Point 3: Line 17, please remove "and so on".

Response 3: "and so on" has been removed.

Point 4: Line 20, Instead of "After that" you could use "Following their work"

Response 4: "Following their work" has been instead of "After that".

Point 5: Line 21, Systems -----> Methods

Line 22, There should be comma after ligands

Line 23, "...Conditions sees necessary [14-17]."

Response 5: The mistakes have been corrected according to the comment.

Point 6: Line 27, "...air stable, AND less toxic property [18-20].

Line 28, Remove "but" and use "However," instead.

Line 29, "Wang Lei and coworkers developed A catalyTIC system."

Line 31, use "in" instead of "with".

Response 6: The mistakes have been corrected according to the comment.

Point 7: Line 37, "...screened AND NiCl2...."

Line 37, "...found to be the best affording the ..."

Line 37, 2a should be 3a, is that right?

Line 38, "...(table1, entry 1)

Response 7: The sentences have been corrected according to the comment.

Point 8: Line 38, remove "Then"

Response 8: "Then" has been removed.                                           

Point 9: Line 40: remove like and use "such as"

Response 9: "like” has been removed.

Point 10: Line 42 and 43 is not necessary to be here. you can remove that.

Scheme 2 is not providing any information. Please remove that.

Response 10: Line 42, 43 and Scheme 2 have been removed.

Point 11: Line 53, has been investigated.

Line 54, remove one and use ligand instead,

Response 11: The incorrect have been changed according to the comment.  

Point 12: Line 65, remove "coupling-cyclize" and use "react"

Line 66, EWG with more steric hinderance such as 4-t-butyl reduces the reactivity and gave 37% yield.

Response 12: The sentences have been corrected according to the comment.

Point 13: Line 79: "... can be achieved instead of "could be get"

Line 80, transform -----> this transformation

line 81, use " was not reported in the literature" instead of ".. has no literature .."

Line 81, Remove Pleasurably and use " Our results shows ..."

Response 13: The words have been changed.

Point 14: Line 98, "... a novel AND simple..."

Response 14: The incorrect have been revised.
